

# Identification of QTL for barley grain size

Junmei Wang[1], Xiaojian Wu[1], Wenhao Yue[1], Chenchen Zhao[2], Jianming Yang[1] and Meixue Zhou[2]

[1] Institute of Crop and Nuclear Technology Utilization, Zhejiang Academy of Agricultural Sciences, Hangzhou, China
[2] Tasmanian Institute of Agriculture, University of Tasmania, Prospect, TAS, Australia

## ABSTRACT

**Background**. Barley grain size is one of the key factors determining storage capacity during grain filling. Large, well-filled grains also have a high malt extract potential. Grain size is a complex quantitative trait and can be easily affected by environmental factors thus the identification of genes controlling the trait and the use of molecular markers linked to the genes in breeding program is the most effective way of improving grain size.

**Methods**. Grain sizes of 188 doubled-haploid (DH) lines derived from the cross of a Japanese malting barley variety (Naso Nijo) and a Chinese feed barley variety (TX9425) were obtained from three different sites in two consecutive years. The average data were used for identifying QTL for grain size.

**Results**. A total of four significant QTL were identified for grain length (GL) and three for grain width (GW). The two major GL QTL are located at similar positions to the QTL for malt extract on 2H and *uzu* gene on 3H, respectively. However, the GL QTL on 2H is more likely a different one from the malt extract QTL as most of the candidate genes are located outside the fine mapped QTL region for malt extract. The GL QTL on 3H is closely linked with *uzu* gene but not due to a pleiotropic effect of *uzu*. The three QTL for grain width on 1H, 2H and 5H, respectively, were located at same position to those for GL.

## INTRODUCTION

Barley (*Hordeum vulgare* L.) is considered to be the first crop domesticated by human ancestors back to 10,000 years ago (*Zohary, Hopf & Weiss, 2012*). It is not only an important crop model for studies on genetics, biochemistry and biology development (*Giraldo et al., 2019*), but is also widely used for animal feeding, malting and brewing, as well as human food (*Zhou, 2010*; *Walker & Panozzo, 2016*; *Wendt et al., 2016*). Barley grain yield is determined by various yield components (*Benbelkacem, Mekni & Rasmusson, 1984*; *Hadjichristodoulou, 1990*; *Peltonen-Sainio et al., 2007*). The sink capacity, a key to grain yield, is a function of the number of grains per unit land area and their potential size (*Bingham et al., 2007*). Barley grain size correlates with not only barley yield (*Xu et al., 2018*) but also grain texture, such as malting quality, starch and protein content (*Walker et al., 2013*; *Yu et al., 2017*), is

Corresponding author
Meixue Zhou, mzhou@utas.edu.au

thus an important end-use quality parameter (*Holopainen et al., 2005*; *Psota et al., 2007*; *Mayolle et al., 2012*). Enhancing grain size is, therefore, one of efficient ways of increasing yield and improving end-use efficiency of barley.

Grain size in rice has attracted more attention than that in barley, with many genes/QTL determining grain sizes having been identified. These include *grain size 5*, *GS5*, which encoding a putative serine carboxypeptidase (*Li et al., 2011*), *qGL3* and *1GW2a* (*Sun et al., 2013*), *GL3.1*, a protein from phosphatase kelch (PPKL) family which influences protein phosphorylation in the spikelet and accelerates cell division (*Qi et al., 2012*), and *GS3* which are composed of four domains that function differently in determining grain sizes (*Fan et al., 2006*; *Mao et al., 2010*). In comparison to rice, there are only limited reports on QTL for barley grain size. From a DH population of a cross between six-rowed and two-rowed barleys, eight QTL were identified for grain length (GL) and nine for grain width (GW) (*Wang et al., 2019*). Among these QTL, the *vrs1(HORVU2Hr1G092290)* was annotated to be the candidate gene for the major QTL on 2H as a homeobox-Leucine Zipper Protein which controls row type and *nud* (*HORVU7Hr1G089930*) that determines the hulled/naked caryopsis morphology is more likely the candidate gene for chromosome 7H QTL hotspot region underlying grain size, though this candidate gene was annotated an ethylene-responsive transcription factor (*Wang et al., 2019*). Five QTL were identified from crosses between a wild barley and two cultivated barley cultivars with two major ones on 3H (20.7–25.6 cM) and 4H (4H 62.9–69.9 cM), respectively (*Zhou et al., 2016*). Eleven grain length QTL were identified from a cross between two Australian malting varieties, Vlamingh and Buloke (*Walker et al., 2013*) but only two major QTL for grain length (qGL2H and qGL5H) were identified from the same cross in a different study (*Watt et al., 2019*). qGL5H has been fine mapped to a 1.7 Mb interval comprising of 23 candidate genes related to grain sizes (*Watt et al., 2019*). By employing the same material, qGL2H was fine mapped to a 140.9 Kb interval and a gene that encodes a MYB transcription factor is likely the candidate for this QTL (*Watt et al., 2020*). Genome-wide association studies (GWAS) have also been successfully used in mapping grain size related QTL in sorghum (*Tao et al., 2020*), rice (*Ma et al., 2019*) and Wild Wheat *Aegilops tauschii* (*Arora et al., 2017*).

In the current study, 188 doubled-haploid (DH) lines derived from the cross between a Japanese malting barley (Naso Nijo) and a Chinese feed barley (TX9425) were grown in eight different environments (three different sites × multiple years). Seven significant QTL for grain size were identified on 1H, 2H, 3H and 5H, respectively. One of the QTL (*QGl.NaTx-2H*) was located at a similar position to a previously reported malt extract QTL (*Wang et al., 2015*). Given that malt extract QTL were once reported to corelate with grain size (*Walker et al., 2013*), we analysed the QTL for grain size using malt extract as covariates. Our results suggested a close linkage between malt extract and grain size, but these two traits are controlled by different genes, which was confirmed by various near iso-genic lines, adding the novelty to our findings. Overall, this study identified new QTL for grain sizes and investigated the relationship between grain size and major quality traits.

## MATERIALS AND METHODS

### Plant materials and field experiments

A doubled haploid (DH) population containing 188 lines derived from the $F_1$ of the barley cross between TX9425 (a Chinese feed barley, two-rowed) and Naso Nijo (a Japanese malting barley, two-rowed) by the anther culture method was employed for identifying QTL determining grain width and length. More details referring to this DH population and their parents can be found in previous studies (*Wang et al., 2015*). All the DH lines and parents were obtained from Tasmanian Institute of Agriculture, University of Tasmania and were grown in Hangzhou (HZ), Zhejiang province, and Yancheng, Jiangsu province (YC) in three successive growing seasons (2006/07 (06), 2007/08 (07) and 2010/11(11)), and Baoshan, Yunnan province (BS) in two continuous growing seasons (2006/07 (06) and 2007/08 (07)). HZ had a slightly higher temperature and more rainfall than YC and BS during the grain filling stage. 150 vigorous seeds of each line or variety were sown in a 2 m row with 0.25 m spacing between rows. All experiments were arranged as a randomized complete block design with three replications. All fields were cultivated with medium fertility, manually weeding and rainfall irrigated. On maturity, grains of each line or variety were harvested for target analysis.

### Phenotypic measurements

At maturity, kernels of plants were bulk-harvested and sun-dried for seed morphological analysis. Grain size traits, grain length (GL, mm), and grain width (GW, mm) were manually measured. For GL estimation, 15 randomly picked kernels from the bulked kernels were lined head to toe horizontally and the total length was estimated using an electronic LCD digital calliper. For GW estimation, instead of lining seeds head to toe, 15 randomly selected seeds were lined side by side and the total length was estimated by the same calliper. The average of three replicated measurements for both grain length and width was recorded for further analysis.

### Statistical analysis

Analysis of variance (ANOVA) was conducted on replicated measurements from eight sites/years using IBM SPSS statistical analysis software (Chicago, USA). Single environmental effect and combined environmental effects on GL and GW were also analysed. Best linear unbiased predictions (BLUPs) for grain size characteristics were calculated using linear mixed models for individual trials and a combined analysis of all field trials known as a multi-environment trial (MET). The simplified model is given by

$$y = X_t + Z_u + e$$

where $y$ is the vector of observations for different grain size characteristics; $X$ is a design matrix associated with a vector of fixed effects $t$; $Z$ is a design matrix associated with a vector of random effects $u$; and e is the vector of residuals that include residual error variance (*Smith et al., 2019*). Trait BLUPs were obtained using linear mixed models using advanced restricted maximum likelihood techniques. The significance in seed width and seed length between high and low malt extract lines was performed with student one-tail $t$-test.

## Genotype analysis

Genomic DNA was extracted from the leaf tissue of three-week old seedlings, based on a modified CTAB method described by *Stein, Herren & Keller (2001)*. DH lines and the two parental varieties were genotyped with DArTSeq (https://www.diversityarrays.com/technology-and-resources/dartseq/). Due to the large number of DNA markers (~30,000 SNP and DArTSeq markers), markers with the same positions or with greater distortion and missing data were removed from the map construction. These markers were combined with previous genotypic data (DArT and SSR markers) and were employed for QTL analysis (*Xu et al., 2012*; *Wang et al., 2015*).

## QTL analysis

The construction of a genetic linkage map was produced as described earlier (*Wang et al., 2015*). The genetic linkage map produced from the TX9425/Naso Nijo DH population using over 2,500 markers and BLUP data of grain size from different years and sites were used for QTL analysis. The software package MapQTL6.0 (*von Korff et al., 2008*) was used to detect QTL which were first analysed by interval mapping (IM). The closest marker at each putative QTL identified using interval mapping was selected as a cofactor and the selected markers were used as genetic background controls in the approximate multiple QTL model (MQM). A logarithm of the odds (LOD) threshold values applied to declare the presence of a QTL were estimated by performing the genome wide permutation tests using at least 1000 permutations of the original data set for each trait, resulting in a 95% LOD threshold of around 3.0 and the walking speed for the genome-wide scan was set at 1 cM. The percentage of variance explained by each QTL ($R^2$) was obtained by using restricted MQM mapping. Graphical representation of linkage groups and QTL was carried out using MapChart 2.2 (*Voorrips, 2002*).

## Candidate gene annotation

To identify candidate genes underlying grain size QTL, we localised the closest marker on the POPseq genetic map of Morex × Barke (*Mascher et al., 2013*). Barley population sequencing data were downloaded following (*Mascher et al., 2013*). The marker primer sequences were used to blast barley databases on http://webblast.ipk-gatersleben.de/barley/ for candidate genes. To be more accurate in candidate genes predication, cloned genes in rice which determine grain sizes were referred to investigate if any barley homolog genes were located within the identified QTL zone in this project. Since QTL zone was identified by the DArT Markers, the physical distance of the QTL zone was then determined by blasting the up and down marker sequence in a barley database (https://webblast.ipk-gatersleben.de/barley_ibsc/). Subsequently, corresponding protein sequences from cloned rice genes (https://funricegenes.github.io/) were used to blast barley homolog genes in the IPK database, where the physical location was identified and further checked with the reported QTL in this project.

**Table 1** Mean and range of grain size traits tested in different environments.

| Trait | Environment | TX | NN | DH | |
|---|---|---|---|---|---|
| | | | | Mean ± SD | Range |
| GL (mm) | HZ07 | 8.14 | 7.87 | 8.16 ± 0.22 | 7.67–8.74 |
| | HZ08 | 8.42 | 8.07 | 8.43 ± 0.26 | 7.645–9.31 |
| | HZ11 | 8.43 | 8.14 | 8.58 ± 0.25 | 7.61–9.31 |
| | YC07 | 8.47 | 8.13 | 8.55 ± 0.28 | 7.71–9.29 |
| | YC08 | 8.54 | 8.23 | 8.68 ± 0.27 | 7.83–9.39 |
| | YC11 | 8.81 | 8.02 | 8.81 ± 0.28 | 7.81–9.46 |
| | BS07 | 8.73 | 8.51 | 8.79 ± 0.28 | 7.98–9.42 |
| | BS08 | 8.57 | 8.35 | 8.75 ± 0.22 | 8.01–9.3 |
| GW (mm) | HZ07 | 3.72 | 3.82 | 3.79 ± 0.08 | 3.55–4.05 |
| | HZ08 | 3.56 | 3.73 | 3.66 ± 0.08 | 3.22–3.88 |
| | HZ11 | 3.68 | 3.75 | 3.71 ± 0.09 | 3.51–4.02 |
| | YC07 | 3.62 | 3.85 | 3.67 ± 0.09 | 3.42–3.96 |
| | YC08 | 3.64 | 3.83 | 3.7 ± 0.08 | 3.48–3.99 |
| | YC11 | 3.77 | 3.87 | 3.78 ± 0.08 | 3.57–4.02 |
| | BS07 | 3.65 | 3.76 | 3.72 ± 0.08 | 3.52–3.94 |
| | BS08 | 3.74 | 3.79 | 3.78 ± 0.18 | 3.49–4.13 |

**Notes.**

SD, standard deviation; TX, a Chinese feed barley variety TX9425; NN, a Japanese malting barley variety Naso Nijo; DH, double-haploid; GL, grain length; GW, grain width; HZ, YC, BS, represent different locations (Hangzhou, Yancheng and Baoshan, respectively) and the number after locations are the year of harvest.

# RESULTS

## Grain size traits for parents and DH lines

We evaluated GL and GW of the DH population and the parents from multiple years/sites. The descriptive mean values of grain length and width for the parents and the double-haploid population in each environment are shown in Table 1. Naso Nijo showed higher values of GW in all environments while TX9425 had higher GL under all environments. Transgressive segregation was found with some DH lines showing higher or lower values than both parents (Table 1). The effects of genotypes, locations, and years were highly significant for the grain size traits (Table 2). For example, one parent TX and DH lines showed generally higher GL in YC11 than trials from other locations. Interactions between genotypes, locations, and years were also significant for these traits.

## QTL analysis for grain length (GL)

Four significant grain length QTL (*QGl.NaTx-1H, QGl.NaTx-2H, QGl.NaTx-3H* and *QGl.NaTx-5H*) were identified using MET-BLUP data from all different environments (Fig. 1, Table 3). These QTL explained 6.8–29.8% of the phenotypic variation. The total phenotypic variation explained by these six QTL was about 70%. Naso Nijo alleles of *QGl.NaTx-1H* and *QGl.NaTx-3H* contributed to longer grain while for *QGl.NaTx-2H* and *QGl.NaTx-5H,* TX9425 alleles contributed to longer grain (Table 3). The most significant QTL, *QGl.NaTx-2H,* was identified on 2H with the closest marker of 3256205S2. This QTL explained 29.8% of the phenotypic variation with a LOD value of 21.95 (Fig. 1, Table

**Table 2** Analysis of variance on grain size traits in DH population lines from Naso Nijo × TX9425 ( *F* values).

| Source of variation | GL | GW |
|---|---|---|
| Block | 8.136** | 3.0* |
| Genotype(G) | 12.53** | 8.3** |
| Location(L) | 1215.95** | 186.89** |
| Year(Y) | 208.15** | 44.5** |
| Y ×L | 109.74** | 483.38** |
| G ×Y | 1.53** | 2.11** |
| G ×L | 1.59** | 2.39** |
| G ×L ×Y | 1.65** | 2.2** |

**Notes.**
*Significant at the 5% level.
**Significant at the 1% level.
GL, grain length; GW, grain width.

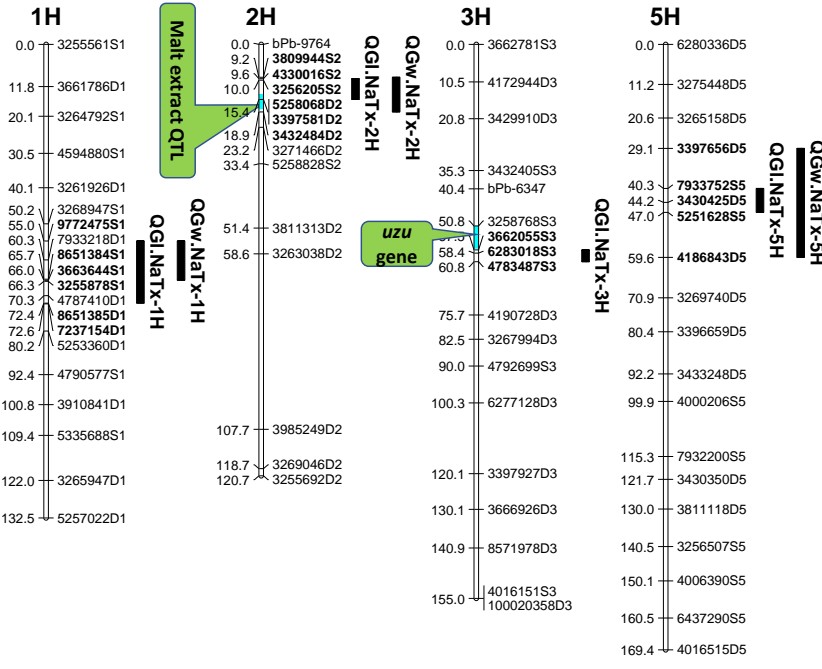

**Figure 1** QTL associated with grain size in barley.

3). Another major QTL *QGl.NaTx-3H.1* with the closest marker of 6283018S3 explained 21.9% of the phenotypic variation. These two major QTL were less affected by growing conditions and were detected in most of the trials (Table S2). The other QTL showed a significant interaction with growing conditions and only detected in some environments, four of eight for the QTL on 1H and two of eight for the QTL on 5H (Table S2).

Wang et al. (2021), *PeerJ*, DOI 10.7717/peerj.11287

Peer J

**Table 3** QTL for barley grain size traits in the DH population of Naso Nijo × TX9425.

| Trait | Linkage group | QTL name | Nearest marker | Position (cM) | Two LOD support intervals | LOD | $R^2$ (%) | Source of positive effect | Additive effect | Malt extract as covariate | *Uzu* Gene as covariate |
|---|---|---|---|---|---|---|---|---|---|---|---|
| GL | 1H | *QGl.NaTx-1H* | 3255878S1 | 66.29 | 54.98–72.38 | 10.64 | 11.9 | NN | 0.073 | NC | NC |
| | 2H | *QGl.NaTx-2H* | 3256205S2 | 10.02 | 9.56–15.44 | 21.95 | 29.8 | TX | −0.111 | 23.0 | NC |
| | 3H | *QGl.NaTx-3H* | 6283018S3 | 58.43 | 57.51–60.84 | 17.46 | 21.9 | NN | 0.259 | NC | 11.6 |
| | 5H | *QGl.NaTx-5H* | 3264393S5 | 47.19 | 40.34–47.01 | 6.25 | 6.2 | TX | 0.044 | NC | NC |
| GW | 1H | *QGw.NaTx-1H* | 4170979D1 | 65.7 | 54.98–66.00 | 4.13 | 9.5 | NN | 0.02 | NC | NC |
| | 2H | *QGw.NaTx-2H* | 5258068D2 | 15.44 | 9.24–18.91 | 7.69 | 18.5 | TX | −0.025 | 15.7 | NC |
| | 5H | *QGw.NaTx-5H* | 3430425D5 | 44.24 | 29.10–59.58 | 3.73 | 8.5 | NN | 0.018 | NC | NC |

**Notes.**

The position is that of the nearest marker; $R^2$ means percentage genetic variance explained by the nearest marker; Two LOD support intervals were used to indicate the 95% confidence intervals (*van Ooijen, 1992*); NC means no significant changes.

### QTL analysis for grain width (GW)

Three QTL (*QGw.NaTx-1H, QGw.NaTx-2H*, and *QGw.NaTx-5H*) were detected for GW based on BLUP from all environments (Table 3). *QGw.NaTx-1H* explained 9.5% of the phenotypic variance, with 4170979D1 being the closest marker and Naso Nijo allele contributing greater grain width. *QGw.NaTx-2H* was located on 2H with the nearest marker of 5258068D2, explaining 17.8% of the phenotypic variation. TX9425 contributed to the wider grain allele. *QGw.NaTx-5H* was located on 5H with the closest marker of 3273028D5, explaining 8.5% of the phenotypic variation. The major QTL *QGw.NaTx-2H* were identified in most of the environments while *QGw.NaTx-1H* and *QGw.NaTx-5H* showed significant interactions with environments, being significant in only three and two environments, respectively. All three QTL were located at similar positions to those for GL.

### QTL analysis for grain length using malt extract as a cofactor

Among the identified QTL for grain size, *QGl.NaTx-2H* and *QGw.NaTx-2H* were located to a similar position of a reported major QTL controlling malt extract using the same population (*Wang et al., 2015*). To confirm if these QTL are conferring to the same gene, QTL analysis for grain size was further conducted using malt extract as a covariate. By doing so, *QGl.NaTx-2H.1* was still significant but the phenotypic variation determined by this QTL reduced from 29.8% to 23.0%, suggesting that GL and malt extract were controlled by different but closely linked genes. Other QTL showed no significant changes in the percentage of phenotypic variation determined when using malt extract as a covariate (Table 3).

### QTL analysis for grain length using *uzu* gene as cofactor

The QTL *QGl.NaTx-3H.1* on 3H was located on a similar position of the *uzu* gene from TX9425 (*Wang et al., 2010*; *Li, Chen & Yan, 2015*; *Chen et al., 2016*) and QTL for awn length (*Chen et al., 2012*). When using awn length as a covariate, phenotypic variation determined by *QGl.NaTx-3H.1* slightly decreased from 21.9% to 16.1% while the percentages variation determined by other QTL were not changed (Table 3), confirming the close linkage between *uzu* and QTL *QGl.NaTx-3H*.

### Correlations between seed size and malt extract

To phenotypically investigate the correlation between seed size (both GL and GW) and malt extract values, we selected representative near isogenic lines (NILs) with high and low malt extract developed from the same cross to compare grain width and grain length. No significant difference in both GL or GW between high and low malt extract NILs ($P > 0.05$) was found, confirming that grain size and malt extract were controlled by different genes.

### Candidate gene predication

Ten genes determining rice grain size and their orthologs in barley were collected in Table S1 . After comparing seven significant QTL identified in this project with rice grain size genes, three barley homolog genes (Table S1), coding carboxypeptidase (1H), cytochrome (5H), mitogen-activated protein (1H), respectively, were identified. No rice homolog genes were identified on 2H and 3H QTL in this study. However, based on literature, several

genes which can affect grain size are located within these QTL regions. Genes related to phytochrmones biosynthesis and cell elongation within these QTL regions were also listed as potential candidate genes (Table S1).

## DISCUSSION

Barley grain sizes positively associated with starch contents which contributes to process performance in human food, animal feed and brewing (*Yu et al., 2017*). Improving grain size is one of the objectives in breeding programs for not only improved quality but also high yielding (*Serrago et al., 2013*). Grain size is a quantitative trait controlled by multiple genes (*Zhou et al., 2016*) and can also be affected by the environment (*Walker et al., 2013*). The identification of QTL and molecular markers linked to grain size is essential for breeders to pyramid different QTL through marker assisted selection. Only a limited number of QTL for grain size have been identified with some larger effect ones on 2H, 3H, 4H and 5H. In this study, we have identified four QTL for GL, and three QTL for GW. Among the four identified significant QTL for GL, *QGl.NaTX-2H* and *QGl.NaTx-3H* were less affected by environments and determined a large percentage (29.8% and 21.9%, respectively) of phenotypic variation (Table 3). A major QTL for GL have been reported on 2H from the cross of Vlaminh and Buloke which is located at 70-80 cM in one report (*Walker et al., 2013*) but at 159–179 cM in another report (*Watt et al., 2019*) which is further fine mapped to a 140.9 Kb interval (*Watt et al., 2020*). The QTL are apparently different from our major QTL on 2H for GL which was located at 10.02 cM (Table 3). The 2H QTL identified in this study determined nearly 30% of the phenotypic variation with the closest marker of 32562045S2 at the position of 10.02 cM. At a similar position (16.3–17.5 cM), a QTL was also found from a cross between a long-kernel wild barley and cultivated barley cultivars but only determined a small proportion (10.4%) of the phenotypic variation (*Zhou et al., 2016*).

The other major GL QTL identified on 3H (*QGl.NaTx-3H*) in this study was in a similar position to *uzu* gene which controls plant dwarfness and has a pleiotropic effect on spike morphology (*Chen et al., 2016*). QTL analysis using *uzu* gene as a covariate indicated that this QTL was not the same but closely linked to *uzu* (Table 3). This QTL has also been reported in previous studies (*Ayoub et al., 2002*; *Zhou et al., 2016*) from populations with no *uzu* gene, confirming that the QTL was not due to a pleiotropic effect of *uzu*. The major QTL for grain size on 2H was not reported before, and the analysis using malt extract as covariate only suggested the linkage of malt extract and grain size, belonging to separate genes/traits that can be selected independently.

QTL alleles determining seed size also tend to determine malt quality. QTL alleles leading to increased variability of kernel size were associated with poor malt quality (*Ayoub et al., 2002*). In our study, the QTL on 2H for GL (*QGl.NaTx-2H*) is located at a similar position to a previously reported QTL for malt extract (*QMe.NaTx-2H*) (*Wang et al., 2015*). To investigate whether these two QTL are the same, we further applied QTL analysis for GL using malt extract as a covariate. Results suggested that these two QTL are independent, instead of a single one gene with pleotropic effect. To further confirm this, we checked

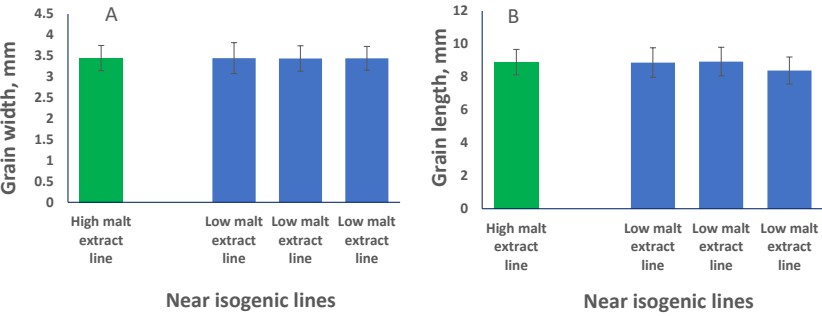

**Figure 2 Grain length (B) and grain width (A) of near isogenic lines.** The pair of NIL were selected from $F_8$ recombinant inbred lines from the cross of TX9425 and Naso Nijo. The markers linked to malt extract was used to select heterozygous individuals and then selfed. Homozygous lines from the next generation ($F_9$) were selected as NIL pairs. These pairs were genotyped with high density markers and evaluated for malt extract. The pair we used here (one line with high malt extract and three lines with low malt extract) showed significant difference in malting quality and the whole genome marker screening showed only differences in the QTL region for malting extract (14 –18 cM, Fig. 1).

several pairs of near isogenic lines (NILs) differing in malt extract QTL. No significant differences were found between lines with high malt extract and those with low malt extract (Fig. 2).

Three barley ortholog genes were found within the identified QTL regions in this study through protein sequence alignment to the cloned grain size genes in rice. Within *QGl.NaTx-1H*, *HRVU.MOREX.r2.1HG0042890* is an ortholog to *OsGS5*, encoding a serine carboxypeptidase and functions as a positive regulator of grain size (*Li et al., 2011*). Another ortholog gene (*HORVU.MOREX.r2.1HG0040860*) of *OsMAPK6* was also identified within this QTL region, encoding mitogen-activated protein kinase 6, determining rice grain size (*Liu et al., 2015*). SMALL GRAIN 1 is another mitogen-activated protein kinases identified in rice, involving regulating rice grain sizes, its homolog gene *HORVU.MOREX.r2.5HG0381450* was identified within *QGl.NaTx-5H* and encode cytochrome, a protein involving cell wall elongation in barley (Table S1). Several candidate genes linked to cell growth and phytohormones also exist in the GL QTL region. ABC transporters play critical roles in plant growth and development, especially for the development of specialized plant cells (*Do, Martinoia & Lee, 2018*) and regulation of root cell growth (*Larsen et al., 2007*). MYB transcription factor is also proposed to be the candidate for GL (*Watt et al., 2020*). It is linked with cell growth and seed production through interacting with plant hormones, playing roles in sperm-cell, stamen development, cotton fibre and even stomatal cell divisions (*Lai et al., 2005*; *Rotman et al., 2005*; *Pu et al., 2008*; *Zhang et al., 2010*). Cytochrome P450 gene, which belongs to CYP78A subfamily, was reported to have a function in seed and fruit size development (*Tian et al., 2016*). Zinc finger proteins could also be good candidates for seed growth as they not only play a strong role in regulating cell growth, but also are critical for chloroplast and palisade cell development, thus affect seed filling and change seed size (*Næsted et al., 2004*). Manipulating ethylene signalling also indicates evidence to improve yield-related traits in crops. Overexpression of

an ethylene response element MHZJ, a membrane protein, promoted grain sizes in rice (*Ma et al., 2013*). Similar findings were also observed in a wheat study, where overexpression of the transcriptional repressor (TaERF3, ethylene response factor), vice versa, decreased grain size and affected 1000-grain weight (*Wang et al., 2020*). IAA and gibberellin play critical roles in regulating seeds size, such that IAA-glucose hydrolase gene *TWG5* determines grain length and yield (*Ishimaru et al., 2013*) and the identified quantitative locus *GW6* controls rice grain size and yield through the gibberellin pathway (*Shi et al., 2020*). Based on these reported genes and their functions in determining grain sizes, we highlighted 66 genes involving the discussed functions in accordance with the identified QTL from this population. Most of those candidate genes are located outside the area for malt extract (Fig. 1).

## CONCLUSIONS

In this study, seven major QTL for grain size were identified. The major one on 2 H (*QGl.NaTx-2H*) is closely linked to the reported QTL for malt extract (*QMe.NaTx-2H*, (*Wang et al., 2015*). The other major QTL on 3H for GL (*QGl.NaTx-3H*) shares a similar position with a reported dwarf gene, *uzu* (*Chen et al., 2016*), but they are two independent genes and control different phenotypes. Therefore, these major QTL can be used in breeding program to improve grain size, independent of malting quality and plant height.

### Funding

This work was supported by the National Natural Science Foundation of China (31671678), the China Agriculture Research System (CARS-5), the Key Research Foundation of Science and Technology Department of Zhejiang Province of China (2016C02050-9) and the Grains Research and Development Corporation (GRDC) of Australia. The funders had no role in study design, data collection and analysis, decision to publish, or preparation of the manuscript.

### Grant Disclosures

The following grant information was disclosed by the authors:
National Natural Science Foundation of China: 31671678.
China Agriculture Research System (CARS-5).
Key Research Foundation of Science and Technology Department of Zhejiang Province of China: 2016C02050-9.
Grains Research and Development Corporation (GRDC) of Australia.

### Competing Interests

The authors declare there are no competing interests.

## Author Contributions

- Junmei Wang conceived and designed the experiments, performed the experiments, analyzed the data, prepared figures and/or tables, authored or reviewed drafts of the paper, and approved the final draft.
- Xiaojian Wu, Wenhao Yue and Jianming Yang performed the experiments, authored or reviewed drafts of the paper, and approved the final draft.
- Chenchen Zhao analyzed the data, prepared figures and/or tables, authored or reviewed drafts of the paper, and approved the final draft.
- Meixue Zhou conceived and designed the experiments, analyzed the data, prepared figures and/or tables, authored or reviewed drafts of the paper, and approved the final draft.

## Data Availability

The barley sequence dataset is available at: https://www.diversityarrays.com/technology-and-resources/sequences/.

Raw data of grain size are available in the Supplemental File.

## Supplemental Information

Supplemental information for this article can be found online at http://dx.doi.org/10.7717/peerj.11287#supplemental-information.

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
