# Peer review of "Identification of QTL for barley grain size"

_PeerJ, doi:10.7717/peerj.11287_

## Round 0.1 · original submission · Major Revisions

Please find the reviewers' comments on your manuscript. Please revise accordingly.

Reviewer 1 ·

Basic reporting

no comment

Experimental design

no comment

Validity of the findings

no comment

Additional comments

The manuscript entitled “Identification of QTL for barley grain size” by Wang et al is focused on dissecting the genetic constitution of grain size in barley. Some interesting results have been obtained, which are valuable for understanding the genetics of grain traits in barley. However, the manuscript has some problems and needs correction before publishing. The author’s argument should be expressed in clear and readable English. There should be no errors of grammar. I have found numerous errors. Therefore, many improvements have to be made before this manuscript could be published in PeerJ.

1. Line 121, “The QTL analyses were based on BLUP from different sites/years”, it reads awkward. Please consider changing it” to “The QTL analyses were based on BLUP data of grain size from different years and sites”.

2. Line 157, Authors mentioned that MET-BLUP data were used in this study. However, they did not perform any methods of data analysis for MET-BLUP in the manuscript. The statistical analysis in Materials and Methods need to be described with more details about MET-BLUP.

3. Line 159, “Naso Nijo contributed longer grain alleles for QGl.NaTx-1H and QGl.NaTx-3H while TX9425 contributed longer grain alleles for QGl.NaTx-2H and QGl.NaTx-5H”, please rewrite this sentence.

4. Line 170, “QGw.NaTx-1H explained 9.5% of the phenotypic variance, with 4170979D1 being the closest marker and Naso Nijo contributing wider grain allele”, please rewrite this sentence.

5. Line 194, the subtitle “Comparison of seed size among different pairs of near isogenic lines differing in malt extract” is too long, please rewrite it.

6. Line 201, “Ten rice genes were identified in determining rice grain sizes (Table S1)” should be changed to “Ten genes determining rice grain size and their orthologs in barley were collected in Table S1”.

7. Line 205, “No rice homolog genes were identified on 2H and 3H QTL. However, genes within these QTL potentially involving phytohormones biosynthesis, cell elongations were also listed as potential candidate genes”, please rewrite this sentence to describe the results of QTL analysis more clear.

8. Line 237-241, “QTL alleles that made seeds larger or rounder also tended to improve malt quality but QTL alleles that increased the variability of the grain size were associated with poor malt quality (Ayoub et al., 2002). In this study, the QTL on 2H for GL is located at a similar position to that for malt extract (Wang et al., 2015). QTL analysis for GL using malt extract as a covariate showed a close linkage between these two genes instead of a pleiotropic effect”. The statement is ambiguous, please rewrite it.

9. Line 244, “Based on the cloned grain size genes in rice, we found three barley ortholog genes to rice within the identified QTL region in this study”, please rewrite this sentence.

10. Line 271, “Based on above knowledge”, please rewrite it.

11. Line 276-278, Several major QTL for grain size were identified. The major one on 2H is linked to the QTL for malt extract but controlled by different genes. The other one on 3H is in a similar position to a dwarf gene, uzu, but not due to a pleiotropic effect of uzu. The statement is ambiguous, please summarize the results in this study more clear and more readable.

12. The specific name of near isogenic lines should be added and showed in Figure 2 and the authors should provide a detailed description in this paper.

13. The word “QTL” should be instead of “QTLs” in many places in the manuscript.

Reviewer 2 ·

Basic reporting

Basic report
1. Professional English. The draft has some issues for text improvement, such as the use of unabbreviated terms right after the abbreviation (line 27 for GW), requirements for rephrasing for some long sentences that are poorly structured, misspelling on Table 3. Sentences in lines 181-183, 222-224, 234-236, 237-239 are unclear and should be rephrased. Also, line 201 is written poorly and looks like plagiarism. I should also mention that English is not my primary language, so I can’t help improve the ms.
2. Introduction and References. It would be more informative to introduce QTL for GL or GW obtained via GWAS, if available.
3. Structure. The structure of the manuscript (ms) is good enough.
4. Figures. Figures of the ms are appropriately provided and given in good quality. Tables 1 and 2 have some abbreviations that should be unabbreviated in notes under Tables.
5. Row data. I didn't find the results of QTL mapping from three different sites and studied years.
Table 2 suggests that location and Year are significantly important for the variance of studied traits. Therefore, it would be informative to learn if those identified QTL were stable in different locations and years or not. Some incomplete information is given in the Results section with reference to Table 3. However, Table 3 has no QTL mapping data based on environment studies.
In the Abstract, the authors stated that grain size might vary in different environments. However, in the Results section, no sufficient data were provided to confirm this claim.

Experimental design

Experimental design
1. Scope of the research.
A. The authors provided results of QTL mapping for key grain parameters, such as grain length and grain width. I wonder why they didn’t include grain weight, which is another important yield-related trait? This way, they may connect selected morphological traits with the yield.
B. It would nice to see if those identified QTL for GL and GW were stable in studied sites and years?

2. Questions and hypothesis. As mentioned above, the authors stated that grain size might vary in different environments. However, in the Results section, no sufficient data were provided to confirm this claim. This is related both to morphological analysis of traits and QTL mapping.
3. High technical and ethical standards. I think there no problems here, except the unfortunate sentence in Line 201.
4. Detailed and informative Methods. Materials and Methods were described sufficiently.

Validity of the findings

The validity of the findings
1. Report on the identification of two novel QTL for grain size is sufficient for the Results' novelty.
2. Provided data were sufficiently robust. However, more statistical analyses, such as a correlation and genotype-environment interaction patterns, might be applied to describe the variation of two traits in studied environments.
3. Speculations. In the Abstract, the authors speculated that grain size might be affected by environmental factors. However, no QTL mapping results for GL and GW supported this statement.
4. Sentences in the Conclusions section should be rephrased.

Reviewer 3 ·

Basic reporting

The manuscript is well written and structured as a standard manuscript on QTL mapping in barley and identification of candidate genes behind QTLs. However, there are some small words missing or incorrect use of English, so proofing of English grammar is recommended.
The introduction give sufficient details and reference for this study, with links to similar studies in rice. This manuscript is a continuation of work carried out by the authors over the last 10 years and could be regarded incremental.
Legends to figure 1 and 2 needs a lot of elaboration, many information is missing, otherwise the figure can not be understood. Statistics in figure 2 is missing. Which isogenic lines were used?

Experimental design

The preparation of the barley DH crossing population was published by the authors some years ago, together with the field trials. It still fulfil general requirements for number of environments, years and replicates, and the grains harvested at that time has been published.
Research question is well define, but narrow and too little for a full publication, short communication seems more appropriate.

Validity of the findings

The findings are interesting, in particular taken together with already published grain and malt quality on the same material.
What would be an interesting addition is to demonstrate the use of identified markers in breeding program.
Or validation of identified candidate genes under the QTLs.
There is no information on nitrogen fertilization of the field trials, as this will effect the protein content and may change the balance between hordein (storage proteins) and albumin. It should be easy to determined total-N on grains and calculate protein-%, do GWAS etc.

Additional comments

see above

---

## Round 0.2 · accepted · Accept

As recommended by the reviewers, I am writing to inform you of the acceptance of your manuscript.

Reviewer 2 ·

Basic reporting

no comment

Experimental design

no comment

Validity of the findings

no comment

Additional comments

In my opinion, the revised version of the manuscript is suitable for publication.

Reviewer 3 ·

Basic reporting

ok, see previous review

Experimental design

ok, see previous review

Validity of the findings

ok, see previous review

Additional comments

I still recommend that grain protein-% and TGW is included in the study.